# Modeling the Spatial Distribution of Swordfish (*Xiphias gladius*) Using Fishery and Remote Sensing Data: Approach and Resolution

**Nan-Jay Su** [1,2,*] **, Chia-Hao Chang** [1,3]**, Ya-Ting Hu** [1]**, Wei-Chuan Chiang** [4] **and Chen-Te Tseng** [3]

1 Department of Environmental Biology and Fisheries Science, National Taiwan Ocean University, Keelung 20224, Taiwan; jhchang02@ntou.edu.tw (C.-H.C.); vannesahu0210@ntou.edu.tw (Y.-T.H.)
2 Center of Excellence for the Oceans, National Taiwan Ocean University, Keelung 20224, Taiwan
3 Fisheries Research Institute, Council of Agriculture, Executive Yuan, Keelung 20246, Taiwan; cttseng@mail.tfrin.gov.tw
4 Eastern Marine Biology Research Center, Fisheries Research Institute, Council of Agriculture, Executive Yuan, Taitung 96143, Taiwan; wcchiang@mail.tfrin.gov.tw
* Correspondence: nanjay@ntou.edu.tw

**Abstract:** Swordfish, *Xiphias gladius* (Linnaeus, 1758), is a commercially important species that is widely distributed throughout three oceans. This species inhabits oceanic waters with preferred environmental ranges and migrates vertically to the surface layer for feeding. However, the spatial distribution pattern and habitat preferences of swordfish have been rarely studied in the Pacific Ocean due to the wide geographic range of this species. This study examined the spatial distribution and preferred ranges of environmental variables for swordfish using two approaches, generalized additive models and habitat suitability index methods, with different spatio-temporal data resolution scales. Results indicated that sea surface temperature is the most important factor determining swordfish spatial distribution. Habitat spatial pattern and preferred environmental ranges, estimated using various modeling approaches, were robust relative to the spatio-temporal data resolution scales. The models were validated by examining the consistency between predictions and untrained actual observations, which all predicted a high relative density of swordfish in the tropical waters of the central Pacific Ocean, with no obvious seasonal movement. Results from this study, based on fishery and remote sensing data with wide spatial coverage, could benefit the conservation and management of fisheries for highly migratory species such as swordfish and tuna.

**Keywords:** spatial pattern; habitat selection; generalized additive models; habitat suitability index; scale effect; data resolution

## 1. Introduction

*Xiphias gladius* (Linnaeus, 1758) is a highly migratory swordfish species that is widely distributed in the open waters of the three oceans outside of polar areas; it is also occasionally found in the coastal waters of the Pacific Ocean [1]. As an oceanic apex predator, this large species grows up to 450 cm in fork length and more than 650 kg in weight [2]. They inhabit waters deeper than 400 m during daytime and migrate vertically to the surface layer of the ocean, shallower than 100 m, for feeding during nighttime [3].

Swordfish are an economically important species, primarily caught as major bycatch by commercial tuna longline fisheries, such as the Japanese and Taiwanese longline fleets in the western and central Pacific Ocean and the Spanish longline fleet in the East Pacific Ocean [4]. Around the Hawaiian Islands,

catch rates of swordfish increases at latitudes up to 35°N–40°N and in the vicinity of temperature fronts when fishing during a full moon [5].

Several studies have demonstrated that environmental effects as well as changes in ocean productivity influence swordfish habitat selection, which results in variability in the swordfish catch rate [6]. For example, Rooker et al. [7] and Hsu et al. [8] demonstrated that the spatial pattern of high catch rates is related to various oceanographic variables, such as sea surface temperature (SST), sea surface salinity (SSS), and chlorophyll-a concentration (CHL). In addition, swordfish were discovered to aggregate in waters with high prey density that corresponded to preferred ranges in mixed layer depth (MLD) and sea surface height (SSH) [9].

Various modeling approaches have been applied to explore the relationships between fishery catch rates and environmental effects; an understanding of these relationships can be further used to develop habitat models. Among those methods, generalized additive models (GAMs) are a commonly used tool for dealing with the complex nonlinear effects of environmental factors. For example, Su et al. [10] used GAMs to examine the environmental influences on the seasonal movement patterns of striped marlin (*Kajikia audax*) in the North Pacific Ocean. The spatial patterns of habitat distribution and optimal ranges of preferred habitat for swordfish were characterized— relative to environmental variability and annual oscillation in climate index—using the habitat suitability index (HSI) in the Atlantic [11] and Indian Oceans [12]. However, similar investigations that cover a large geographical area are lacking for swordfish in the Pacific Ocean.

The objectives of this study were to examine the relationships between environmental factors and the habitat preference of swordfish in the Pacific Ocean using fishery catch and effort data as well as multi-sensor satellite-based remotely sensed measurements of environmental variables; this allowed us to identify the habitat characteristics of the areas with highest relative abundance. We applied two commonly used methods (GAM and HSI) to examine this large data set covering a large geographical area in the Pacific Ocean. Because the scale effect may affect the performance of species distribution models [13,14], two spatio-temporal scales for data resolution (monthly 5° × 5° and weekly 1° × 1° grids) were used to examine the scale effect on model predictive ability.

## 2. Materials and Methods

### 2.1. Fishery Data

Catch and effort data on the Taiwanese distant-water tuna longline fishery in the Pacific Ocean for 2009–2017 were obtained from the Oversea Fisheries Development Council (OFDC) of Taiwan. The data included information on swordfish catch, relevant fishing effort, operation date, and operation location (in latitude and longitude). Catch data were in terms of the number of fish caught in the longline fisheries, whereas effort was in terms of the number of hooks employed by these fleets. We defined the catch rate for swordfish as the number of fish caught per 1000 hooks. To examine the scale effect from data resolution on modeling the habitat spatial pattern, fishery data were segmented into monthly 5° × 5° (coarse) and weekly 1° × 1° (fine) grids for analysis.

### 2.2. Environmental Data

The environmental variables, which were independent explanatory variables, were SSH, SSS, SST, MLD, CHL, and lunar phase (Lunar). The information on the lunar effect was available at the fine temporal resolution of a day. Data on these variables were averaged into both spatial and temporal resolution scales to match the fishery data, and were obtained from the following sources.

(1) SSH, SSS, SST, and MLD data for 2009–2017 were obtained from the HYCOM (Naval Research Laboratory at Stennis Space Center; http://www.hycom.org/) [15,16].
(2) Daily CHL data for 2009–2017 at the spatial resolution of 9 km were obtained from the MODIS-Aqua (NASA Goddard Space Flight Center; http://oceancolor.gsfc.nasa.gov/).

(3)      Daily lunar phase data for 2009–2017 were obtained from the US Navy's Fraction of the Moon Illuminated data set (http://aa.usno.navy.mil/data/docs/MoonFraction.php). Values between 0 (new moon) and 1 (full moon) were used to represent the lunar effect in this study.

*2.3. Modeling Approaches for Spatial Distribution*

GAMs are commonly used to model the relationships between environmental variables and catch rates and have been applied in swordfish research [5]. GAMs are extensions of generalized linear models, which are often used to predict species spatial distributions because they deal well with the non-linear relationships between covariates and the response variable [17]. To handle nonlinearity, an underlying assumption of a GAM is that in addition to being smooth, functions of the predictors can be additive [18].

Six environmental covariates were considered for inclusion in the model, namely SSH, SSS, SST, CHL, MLD, and Lunar, as well as four additional spatio-temporal variables, namely month/week, latitude, longitude, and the interaction between latitude and longitude. The full model for analyzing the catch rate data of swordfish is represented as follows:

$$g(\ln CPUE + c) = s(Lunar) + s(SSH) + s(SSS) + s(SST) + s(CHL) + s(MLD) \\ + s(Latitude) + s(Longitude) + s(Interaction) + Month/Week \tag{1}$$

where *g* is the link function, and *s*( ) is a smoothing function for each of the explanatory covariates. CPUE is the log-transformed swordfish catch rate, which led to more normally distributed residuals, with 0.01 added to avoid log-transformation problems [19]. A periodic smoother was used for the month and week effect to ensure this effect was continuous with respect to year [10,18].

A step-wise GAM was used to identify the effects of the explanatory factors on catch rates. As the environmental covariates and spatial and temporal variables were added, the GAM was evaluated in terms of deviance explained by the model, change in the Akaike information criterion (AIC), and the $\chi^2$ test, to determine the final model for prediction. All predictor variables were treated as continuous. Diagnostic analysis (i.e., the distribution of residuals and quantile–quantile plots) was used to evaluate the model fits and assumption of a lognormal error distribution.

The HSI method was used as an alternative approach to model the catch rate of swordfish, as an indicator of habitat preferences. Based on the frequency distribution of fishing effort at each variable level, the suitability index (SI) values were estimated for each environmental valuable using smooth curves fitted to the observed catch rate data; the estimated values were for the mid-point of each interval over the range of environmental variables. The SI values ranged from 0 to 1 for each environmental factor, where 0 and 1 represent the least-suitable and most-suitable habitat conditions, respectively.

The SI threshold was set at 0.7; SI values > 0.7 indicated an optimal habitat condition. The SI models for each environmental variable were combined to develop an empirical HSI model. Two widely used approaches were applied in this study: the arithmetic mean model (AMM) [20] and the geometric mean model (GMM) [21], which have the following general forms.

$$HSI_{AMM} = \left(\sum_{i=1}^{n} SI_i\right)/n \tag{2}$$

$$HSI_{GMM} = \left(\prod_{i=1}^{n}\right)^{1/n} \tag{3}$$

where n is the number of SI models considered, and $SI_i$ is the SI model for environmental variable i. The final HSI models for the two data resolution scales were evaluated by comparing the predictions with data using linear regression models. By then including each environmental variable in the model, we determined the final HSI model. This final model exhibited the highest consistency, as indicated by it having a linear regression slope closest to 1, intercept closest to 0, and the highest coefficient of determination ($R^2$).

*2.4. Model Validation and Accuracy Assessment*

For both modeling approaches and data resolution scales, the GAM and HSI models were fitted for 2009–2015 with fishery and remote sensing data. We evaluated model performance by predicting the spatial maps of habitat distribution using 2016 and 2017 environmental data and overlapping the predictions on the actual untrained catch rate data for the same period to validate the agreement between prediction and observation. To further examine consistency, we classified the model predictions, as well as catch rate data, based on the percentage into five groups ([0, 0.2]; [0.2, 0.4]; [0.4, 0.6]; [0.6, 0.8]; [0.8, 1.0]). The closest agreement was obtained when the data and predicted values were assigned to the identical group. Maps were drawn to visualize spatial differences by level for both modeling approaches (GAM and HSI) at fine resolution.

## 3. Results

The longline fleets operate throughout the Pacific Ocean, with high fishing effort in tropical areas of the central Pacific between 15°N and 15°S. High catch rates of swordfish occurred in the equatorial region for all seasons; the catch rates were lower in temperate waters of the North and South Pacific Ocean (results not shown). For the GAM analysis, the residuals from the lognormal distribution conformed well to its assumptions based on the quantile–quantile plots: the log-spaced residuals appeared normally distributed (Figure 1) and independent of the values for all of the environmental variables used in the study (Figure 2). No heteroscedasticity was noted from the plots of residuals against the values of each environmental variable for the different data resolution scales (Figure 2).

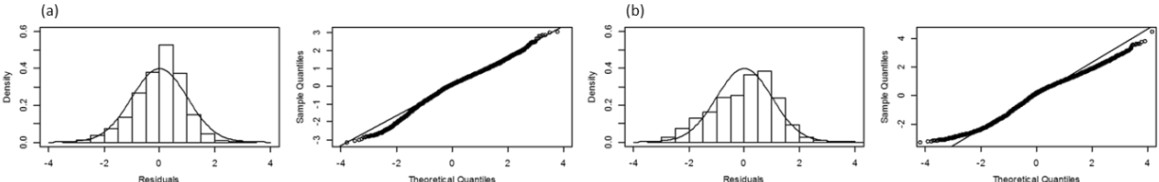

**Figure 1.** Residual distributions and quantile–quantile plots for diagnostic analysis of the final generalized additive models (GAM) with prediction variables (presented in Table 1) based on fishery and remote sensing data aggregated into (**a**) monthly 5° × 5° and (**b**) weekly 1° × 1° grids.

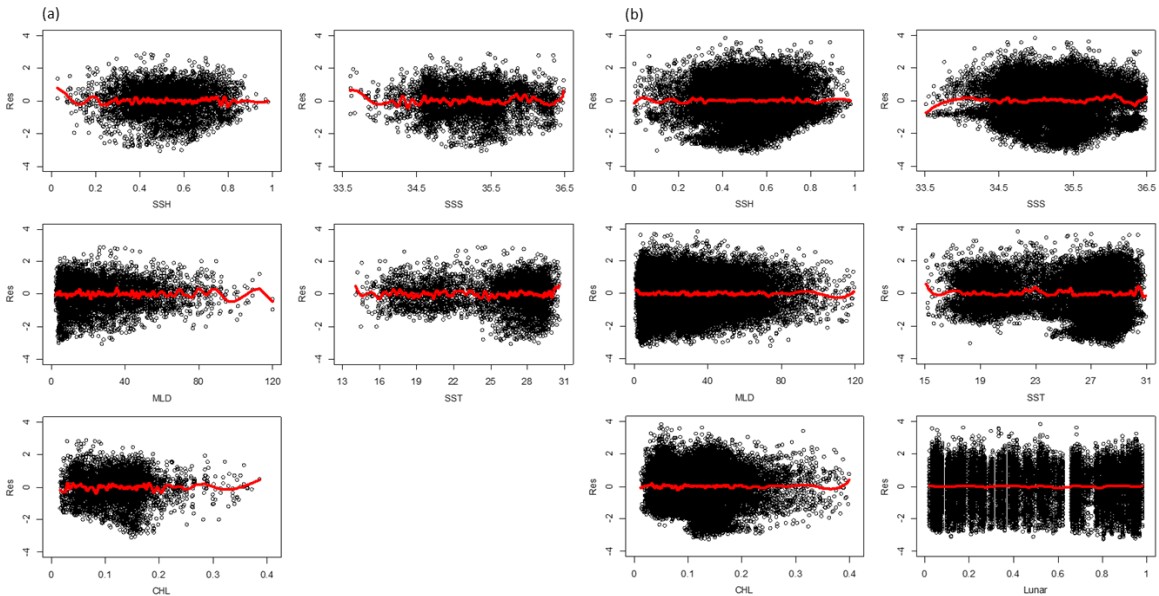

**Figure 2.** Scatterplots of residuals for environmental factors in the final GAM with variables listed in Table 1 based on fishery and remote sensing data aggregated into (**a**) monthly 5° × 5° and (**b**) weekly 1° × 1° grids. The red solid lines in each panel represent the pattern of residuals fitted using a spline function for each environmental variable.

**Table 1.** Deviance, P(χ²), R², and Akaike information criterion (AIC) values for the GAMs selected to analyze fishery data and remote sensing oceanographic variables based on data aggregated into (**a**) monthly 5° × 5° and (**b**) weekly 1° × 1° grids for swordfish in the Pacific Ocean.

| | Residual Deviance | Deviance Explained | P($\chi^2$) | $R^2$ | AIC |
|---|---|---|---|---|---|
| (a) Monthly 5° × 5° Grid | | | | | |
| NULL | 6940 | | | | 17,435 |
| +SSH | 6605 | 334 | <0.001 | 0.048 | 17,163 |
| +SSS | 6289 | 316 | <0.001 | 0.094 | 16,884 |
| +CHL | 5786 | 503 | <0.001 | 0.166 | 16,415 |
| +SST | 5049 | 737 | <0.001 | 0.272 | 15,631 |
| +MLD | 4996 | 53 | <0.001 | 0.280 | 15,580 |
| +Latitude | 4827 | 169 | <0.001 | 0.304 | 15,392 |
| +Longitude | 4797 | 30 | <0.001 | 0.309 | 15,362 |
| +Interaction | 4510 | 286 | <0.001 | 0.350 | 15,029 |
| +Month | 4472 | 39 | <0.001 | 0.356 | 14,986 |
| (b) Weekly 1° × 1° Grid | | | | | |
| NULL | 50,819 | | | | 103,128 |
| +Lunar | 50,294 | 524 | <0.001 | 0.010 | 102,815 |
| +SSH | 48,371 | 1923 | <0.001 | 0.048 | 101,618 |
| +SSS | 46,674 | 1698 | <0.001 | 0.082 | 100,520 |
| +CHL | 45,242 | 1432 | <0.001 | 0.110 | 99,564 |
| +SST | 41,723 | 3519 | <0.001 | 0.179 | 97,068 |
| +MLD | 41,505 | 218 | <0.001 | 0.183 | 96,914 |
| +Latitude | 40,353 | 1152 | <0.001 | 0.206 | 96,051 |
| +Longitude | 40,020 | 333 | <0.001 | 0.212 | 95,801 |
| +Interaction | 38,026 | 1994 | <0.001 | 0.252 | 94,247 |
| +Week | 37,757 | 269 | <0.001 | 0.257 | 94,040 |

All oceanographic variables, spatial and temporal effects, and the interaction term between latitude and longitude included in the GAM analysis were highly significant, as indicated by the χ² tests and the AIC between those models that differed by one covariate (Table 1). The GAMs explained

35.6% and 25.7% of the total deviance for the coarse- and fine-resolution models, respectively (Table 1). The most important environmental factor affecting the swordfish catch rates was SST, which accounted for 29.9% and 26.9% of the explained deviance for the coarse and fine resolutions, respectively. SSS, SSH, and CHL explained 46.7% and 38.7% of the explained deviance for the two final selected GAMs for the coarse and fine resolutions, respectively (Table 1).

The predicted relative densities of swordfish in the Pacific Ocean were plotted against each level of the environmental covariate (Figure 3). Except for CHL, all environmental effects had similar patterns for both coarse and fine resolutions. The effect of SST on the catch rates of swordfish was evident, with a decreasing trend from cold to warm temperatures. In general, the effects of environmental variables exhibited clear patterns, with high catch rates within preferred habitat ranges (Figure 3). When the fine grid was used, high relative densities of swordfish were highly related to the change of the lunar phase (Figure 3).

The observed ranges of the environmental variables for swordfish in the Pacific Ocean are shown in Figure 4. The estimated SIs corresponded well to the observed values for all the environmental variables. The preferred habitat ranges of swordfish were almost identical, regardless of the scale of data resolution used to model habitat suitability (Figure 4). For both cases, swordfish was caught in waters with optimal environmental ranges (SI ≥ 0.7) falling in the middle of the observed extent for SSH, SSS, and CHL. By contrast, the optimal range of MLD for swordfish was shallower than 30 m and was higher than 27 °C for SST (Figure 4). All of the environmental variables were included in the empirical HSI models (AMM and GMM) for both coarse and fine resolutions according to the linear regression analysis (Table 2).

**Table 2.** Results for the linear regression analysis used to evaluate the habitat suitability index (HSI) models, which were developed by adding environmental variables one at a time into the model based on data aggregated into (**a**) monthly 5° × 5° and (**b**) weekly 1° × 1° grids.

|  | Variable | Slope | Intercept | $R^2$ |
|---|---|---|---|---|
| **(a) Monthly 5° × 5° Grid** | | | | |
| AMM | +SST | 0.492 | 0.126 | 0.869 |
|  | +CHL | 0.670 | −0.015 | 0.859 |
|  | +SSH | 0.730 | −0.084 | 0.815 |
|  | +SSS | 0.872 | −0.146 | 0.829 |
|  | +MLD | 1.306 | −0.312 | 0.918 |
| GMM | +SST | 0.492 | 0.126 | 0.869 |
|  | +CHL | 0.657 | 0.018 | 0.902 |
|  | +SSH | 0.667 | −0.007 | 0.881 |
|  | +SSS | 0.740 | −0.016 | 0.926 |
|  | +MLD | 1.019 | −0.039 | 0.944 |
| **(b) Weekly 1° × 1° Grid** | | | | |
| AMM | +Lunar | 0.228 | 0.420 | 0.510 |
|  | +CHL | 0.636 | 0.190 | 0.980 |
|  | +SST | 0.859 | 0.017 | 0.886 |
|  | +SSH | 1.012 | −0.093 | 0.918 |
|  | +SSS | 1.040 | −0.109 | 0.835 |
|  | +MLD | 1.120 | −0.108 | 0.841 |
| GMM | +Lunar | 0.228 | 0.420 | 0.510 |
|  | +CHL | 0.447 | 0.328 | 0.832 |
|  | +SST | 0.542 | 0.275 | 0.698 |
|  | +SSH | 0.687 | 0.193 | 0.837 |
|  | +SSS | 0.823 | 0.129 | 0.957 |
|  | +MLD | 0.930 | 0.122 | 0.945 |

The distributions of nominal and predicted catch rates of swordfish using GAMs showed similar spatial patterns for all seasons, regardless of the resolution used (Figure 5). Similar patterns were found

with habitat SI models used to predict the spatial distribution of swordfish (Figure 6). Both nominal and predicted catch rates were high between latitudes 15°N and 15°S (the equatorial area). However, the two modeling approaches slightly differed in their spatial predictions for waters near New Zealand and the central East Pacific Ocean, particularly for the first half of the year (Figures 5 and 6).

For model validation, the distributions of high relative density of swordfish were predicted using the final GAM and HSI models with 2016 and 2017 remote sensing data, which overlapped well against the actual high catch rates (Figures 7 and 8). In the first half of the year, high relative densities of swordfish, which accorded with predictions from the GAMs, occurred in waters near New Zealand in the southwestern Pacific Ocean (Figure 7). By contrast, the HSI model predicted a high density of swordfish in the central East Pacific Ocean between 5°N and 5°S (Figure 8).

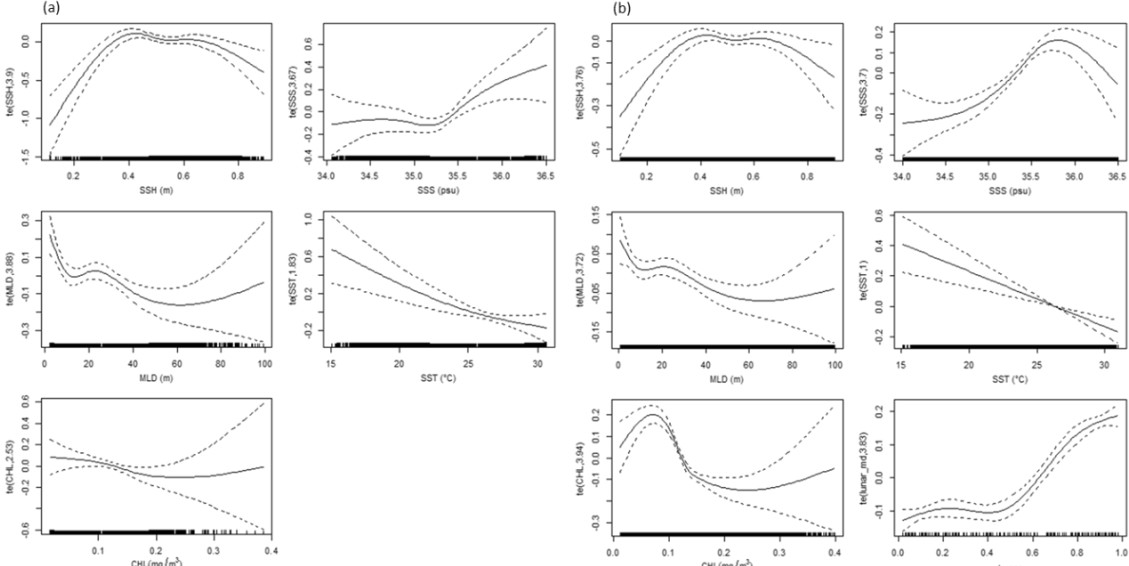

**Figure 3.** Partial response plots in log-space for the environmental effects in the final GAM presented in Table 1 based on fishery and remote sensing data aggregated into (**a**) monthly 5° × 5° and (**b**) weekly 1° × 1° grids. The relative density of data points is marked by the "rug" on the *x*-axis. The "te" sign on the *y*-axis indicates the smooth function used to fit the models, with the degrees of freedom in parentheses. Dashed lines represent the 95% confidence intervals.

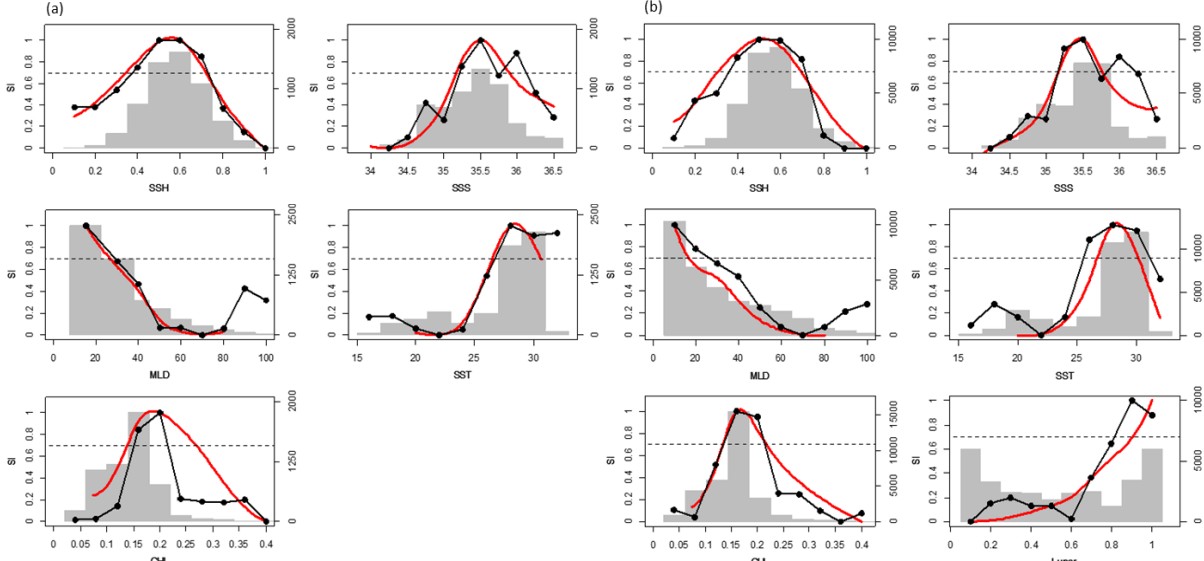

**Figure 4.** Observed (lines with points) and model-predicted (lines without points) suitability indices (SIs) for each environmental effect in the HSI models based on data aggregated into (**a**) monthly 5° × 5° and (**b**) weekly 1° × 1° grids. The number of data points for each environmental variable is presented as a bar chart (right *y*-axis).

The differences between nominal and predicted values from the models were considered small (the blue and green points in Figure 9), although large discrepancies (orange and red points) occurred occasionally in the GAM and HSI approaches for the two grids. Evaluated against actual swordfish catch rate data during 2016 and 2017, HSI models correctly predicted 62.5% and 62.7% of the spatial grids for the coarse and fine grids, respectively. In contrast to the HSI models, GAMs were more accurate: 81.2% and 76.8% of spatial grids were correctly predicted for the coarse and fine grids, respectively.

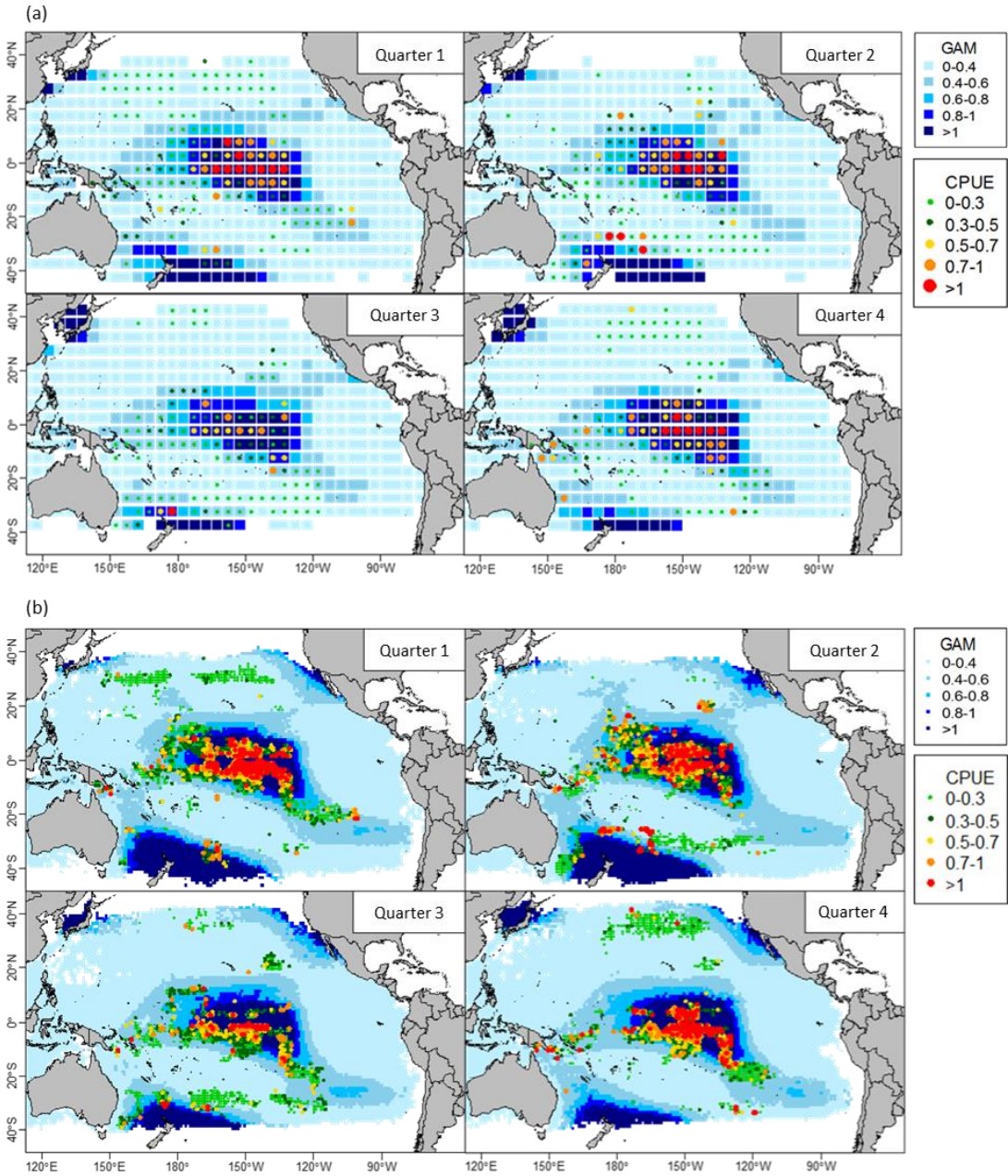

**Figure 5.** Distributions of observed nominal catch rates of swordfish for 2009–2015 overlapping on the maps of predicted relative densities using the final GAM with variables listed in Table 1 based on fishery and satellite-based remotely sensed environmental data aggregated into (**a**) monthly 5° × 5° and (**b**) weekly 1° × 1° grids.

## 4. Discussion

Spatial modeling for delineating hotspots of suitable habitats of target or bycatch species is crucial for fisheries management. In particular, documentation, environmental effects, and techniques for modeling and subsequent mapping are foundational to the use of spatial management tools [22]. In this study, we determined the optimal habitat ranges for swordfish in the Pacific Ocean using alternative modeling approaches (i.e., GAM and HSI) of several important key environmental variables, accounting for the scale effect of data resolution when developing the model. Because highly migratory species are widely distributed, deeper investigation into modeling approaches and data resolution can improve the reliability of model predictions [23,24]. This aids habitat model development by reducing

the dependence between the data points, which reduces the likelihood that the degrees of freedom are overinflated [5].

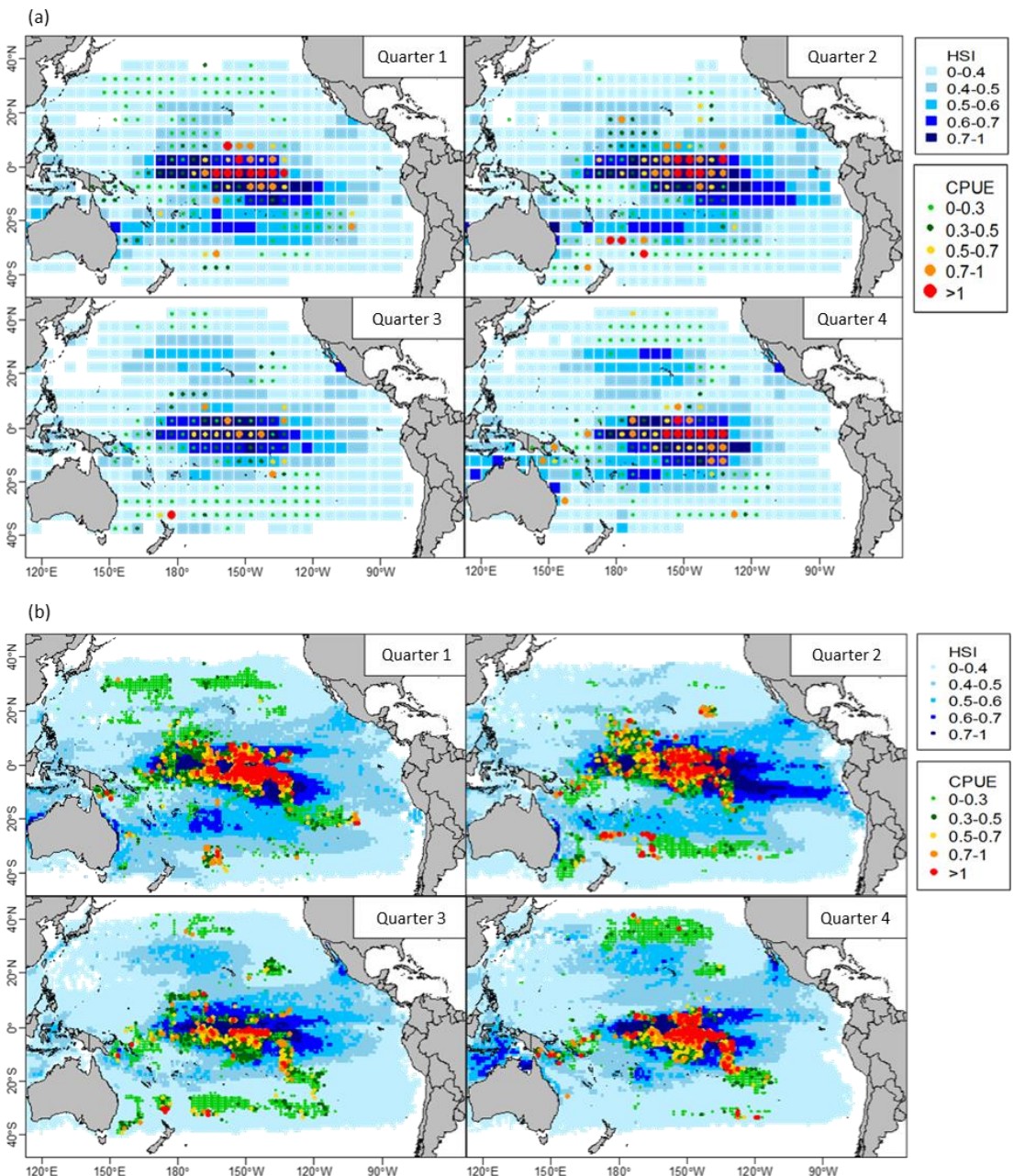

**Figure 6.** Similar to Figure 5, except that HSI models were used for prediction. (**a**) monthly 5° × 5° grids; (**b**) weekly 1° × 1° grids.

Despite the lunar effect, which was only incorporated into the fine-resolution model, the results indicated that scale had negligible effects when modeling the habitat ranges of widely distributed species such as swordfish (see Figures 3 and 4 for comparison). This is consistent with conclusions drawn by Bigelow et al. [5]; SST at different spatio-temporal scales was noted to have similar effects on the catch rates of swordfish in waters near the Hawaiian Islands. However, as demonstrated in the fine-resolution GAM analysis in this study, CHL exhibited a clearer pattern at levels < 0.2 mg/m$^3$, which serves as an important indicator for detecting the transition zone in the Pacific Ocean [25,26]. This result could be related to local variations in CHL due to upwelling and the local topography [27].

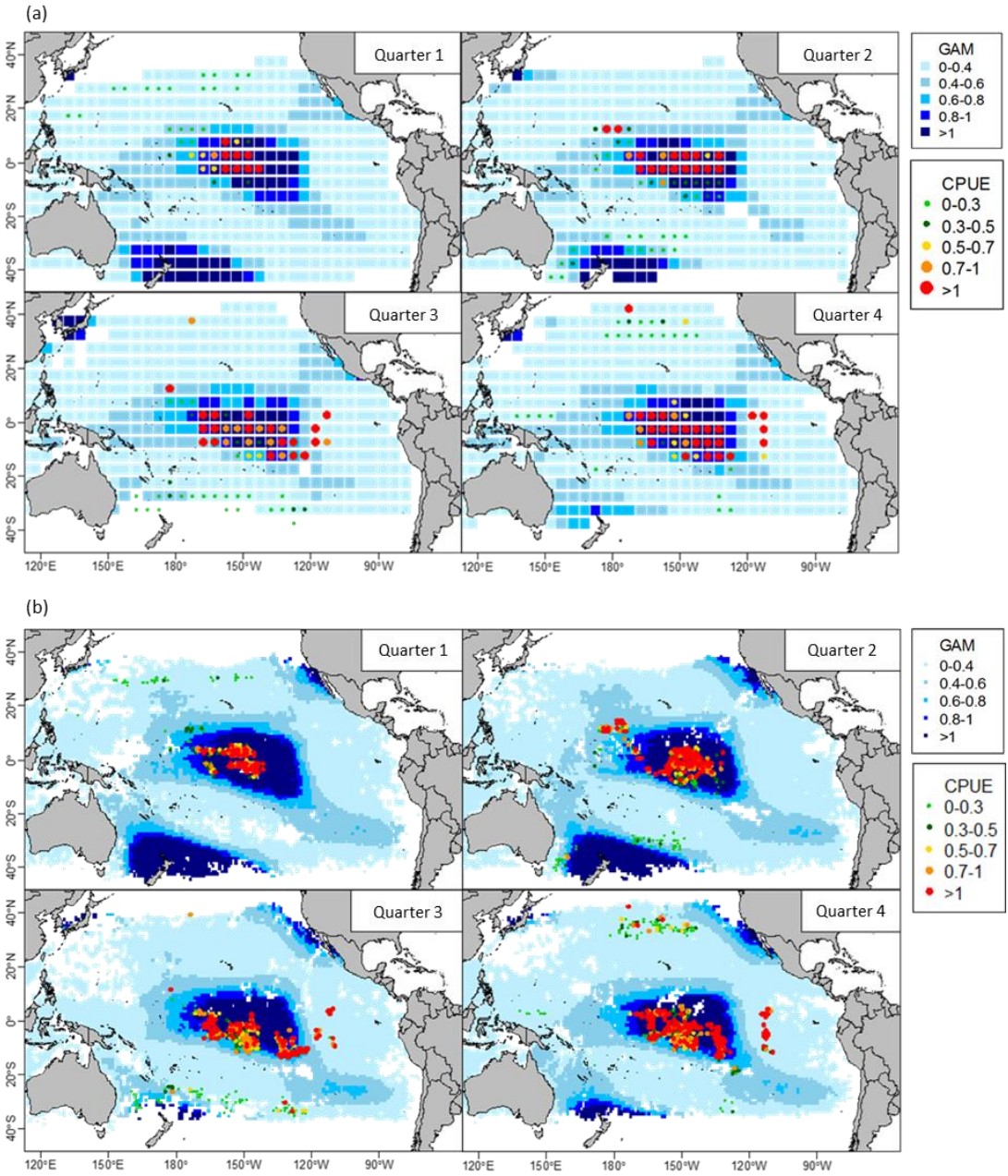

**Figure 7.** Distributions of swordfish catch rates predicted using the final GAM with variables listed in Table 1 and overlaid with nominal 2016 and 2017 observations for validation based on fishery and remote sensing data aggregated at (**a**) monthly 5° × 5° and (**b**) weekly 1° × 1° grids. CPUE represents observed catch rates of swordfish for 2016–2017.

Among the remotely sensed variables considered in this study, SST was the most important factor in determining the spatial pattern of swordfish habitats in the Pacific Ocean (Table 1). Swordfish are considered a pelagic tropical species, and as is typical with this species, no strong seasonal variations and movement in preferred habitats were noted (Figures 5 and 6). Although both modeling approaches furnished similar predictions, the GAM predicted a high relative density of swordfish in waters near New Zealand. This is consistent with prior results [28] that indicated this region had a major fishing ground. Without sufficient fishery data from the southwestern Pacific Ocean and the support of the spatial determinant (latitude and longitude) in the GAM, the empirical HSI models exhibited difficulty predicting regional high-density data within certain habitat ranges [24]. This explained why the HSI

models predicted fewer spatial grids correctly; the models did not predict a high relative density in the southwestern Pacific Ocean.

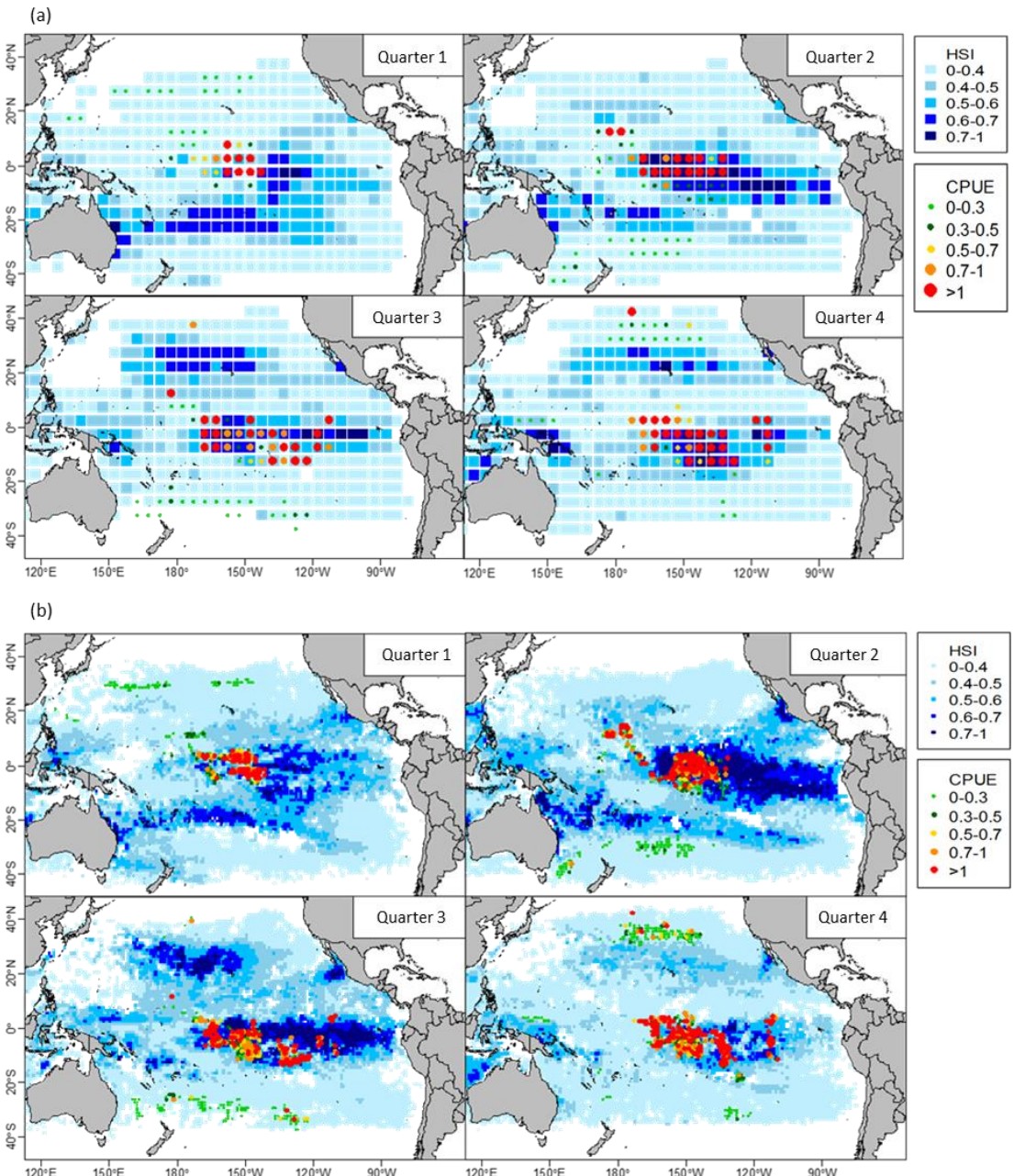

**Figure 8.** Similar to Figure 7, but HSI models were used for prediction. (**a**) monthly 5° × 5° grids; (**b**) weekly 1° × 1° grids.

Both modeling approaches at the fine resolution revealed that the catch rates of swordfish increased nearly linearly with the lunar phase (Figures 3 and 4). This is consistent with previous studies, which have discovered that the highest catch rate of swordfish occurs during the full moon; these studies employed tagging experiments [3] and an analysis of fishery and oceanographic variables [5]. Swordfish migrate vertically from deeper waters to the surface layer at night for feeding, which forms a diel pattern; moreover, because swordfish is a visual top predator, moonlight illumination improves foraging efficiency [13,29]. However, few studies have been conducted to examine the vertical habitat use of swordfish with respect to subsurface oceanographic variables. This limits our understanding

of the relationship between swordfish population distribution and the vertical thermal structure of ocean layers.

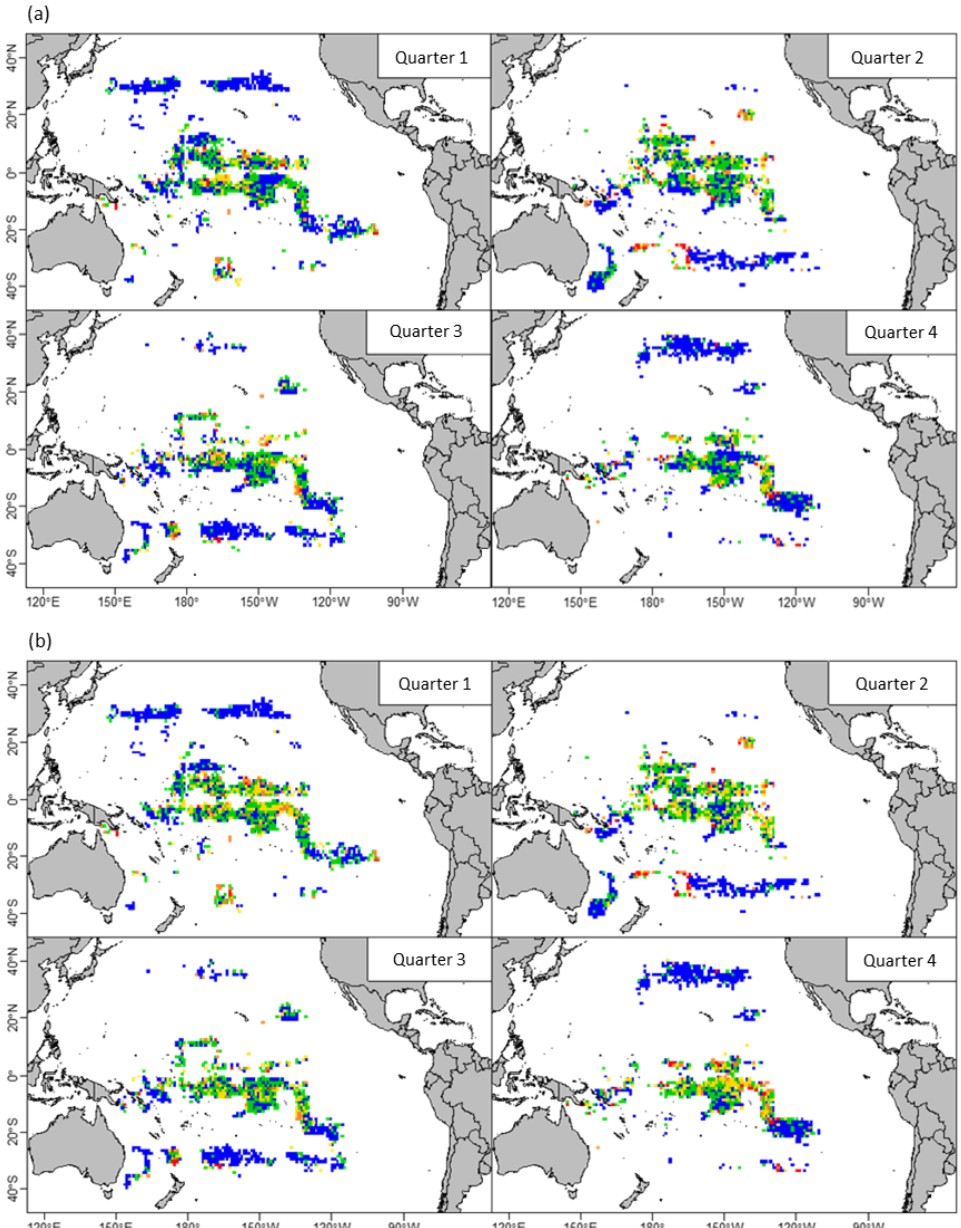

**Figure 9.** Distributions indicating differences by level between the nominal observations and predicted values using (**a**) the final GAM with variables listed in Table 1 and (**b**) HSI models with variables listed in Table 2 based on fishery and remote sensing data aggregated at the weekly 1° × 1° resolution. Small differences are marked as blue and green points, and large differences are marked as yellow, orange and red points.

In the present study, most model outputs of both model-fitting methods, over different data resolution scales for the swordfish habitat distribution, were highly consistent and accurate (Figures 7–9, Table 2). However, the two modeling approaches yielded contradictory predictions of the effect of SST on the swordfish catch rate (Figures 3 and 4). This difference was as expected because SST is highly correlated with latitude, and the HSI approach cannot deal well with the collinearity between these two effects. By contrast, the latitude effect, the longitude effect, and the interaction between the two were included in the GAM analysis, which improves the identification of SST's effect from the spatial factors.

However, because spatial variables are time-invariant arbitrary units on grids that describe geographic location, other important oceanographic and biological variables that relate to the spatial dynamics of swordfish behavior (e.g., feeding and spawning migration) should be included in future analyses.

A major difference between the HSI and GAM approaches was how the environmental variables in the empirical HSI model were weighted. We selected an equal-weighting method for the HSI approach and then further validated the results by comparing the predictions with data sets that were not used to train the model (which were data for 2016 and 2017). Fairly consistent results between predictions and observations and a high success rate were obtained, even though the testing data were changed to any two different years from 2009–2017, making the results range from 56.4% to 82.1%. In addition, the model performance and predictions can be further tested by applying alternative methods such as the area under the curve (AUC), as conducted by Bellamy et al. [23] and Evcin et al. [30].

## 5. Conclusions

The outcomes and predictions of the models, such as the spatial distributions of swordfish and preferred habitats, were robust to both coarse and fine resolutions used in the study (Figures 3–6). However, the lunar effect was identified only at a fine resolution, despite being widely considered a definite factor that increases the catch rate of swordfish [5]. The environmental determinants of swordfish spatial distribution differed between the two modeling approaches. How the SST effect is treated (Figures 3 and 4), for example, may lead to inconsistent conclusions when determining fishing grounds based on a set of remotely sensed environmental variables. Thus, we suggest the use of alternative multi-model approaches to examine the uncertainty resulting from the use of a given modeling approach; results from such multi-model approaches can be used in an ensemble analysis to yield better spatial predictions of a fish species' preferred habitat [18].

Species distribution models have been used to integrate environmental information with species occurrence or abundance data in models or statistical relationships. This study underscores the importance of using alternative modeling approaches to better predict a species' geographic range and determine its core habitat, thus aiding management and conservation. For example, with respect to simulated species occurrence data, the spatially explicit habitat modeling approach has been found to have better performance in predicting species distribution relative to non-spatial methods, such as the boosted regression tree (BRT) and MaxEnt methods [31], even in cases of extreme events [32] and multi-scale problems [23]. This approach can be used in future research.

**Author Contributions:** All authors contributed to this study; conceptualization and methodology, N.-J.S. and W.-C.C.; data analysis, visualization, and writing (original draft preparation), N.-J.S., C.-H.C. and Y.-T.H.; writing (review and editing), W.-C.C. and C.-T.T. All authors have read and agreed to the published version of the manuscript.

**Funding:** This study was funded in part by the Central Weather Bureau and the Ministry of Science and Technology through research grants to the projects (1072053B; 1082045C; MOST106-2611-M-019-010; MOST107-2611-M-019-011).

**Acknowledgments:** We thank the valuable comments and suggestions from the three anonymous reviewers and the editor, as well as the team members of the Central Weather Bureau (CWB), the Fisheries Agency (FA), and the Overseas Fisheries Development Council (OFDC) of Taiwan for their assistance in data preparation.

**Conflicts of Interest:** The authors declare no conflict of interest.

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
