# Peer review of "Modeling the Spatial Distribution of Swordfish (Xiphias gladius) Using Fishery and Remote Sensing Data: Approach and Resolution"

_remotesensing, doi:10.3390/rs12060947_

Round 1

Reviewer 1 Report

I have only two major concerns with the manuscript in the current state. The first concerns Table 1 where the percentage of deviance explained by the different covariates do not seem to match the text - please double check the values in the table against the text in lines 183 -184. Perhaps an earlier version of the model results (rather than the best fit final GAM) were included in the table.

My second major concern is distinguishing P.CPUE vs CPUE in the fine resolution figures 5 - 8 where the square and round symbols can not be clearly separated and the colors are the same e.g. for P.CPUE in the 0.8 to >1 relative density range compared to CPUE in the 0.3 to 0.5 range (both dark blue and easily distinguished by the shape in the coarse resolution figures.

I would like the authors to discuss the differences in SST between the two models before the conclusions since I am curious about these differences and think other readers will be as well. I am glad they were mentioned in the conclusions but think perhaps that should be discussed earlier as well.

The manuscript is a bit wordy and some of the description of the model results are repeated too frequently e.g. there is no need to repeatedly say that there are two different spatial resolutions used .... but this was a minor concern.

My other minor comments and suggestions are included in the attached and edited manuscript.

Author Response

Point 1: I have only two major concerns with the manuscript in the current state. The first concerns Table 1 where the percentage of deviance explained by the different covariates do not seem to match the text - please double check the values in the table against the text in lines 183 -184. Perhaps an earlier version of the model results (rather than the best fit final GAM) were included in the table.

Response 1: We have updated the values and revised the text (Line 495-496).

“The most important environmental factor affecting the swordfish catch rates was SST, which accounted for 29.9% and 26.9% of the explained deviance for the coarse and fine resolutions”

Point 2: My second major concern is distinguishing P.CPUE vs CPUE in the fine resolution figures 5 - 8 where the square and round symbols can not be clearly separated and the colors are the same e.g. for P.CPUE in the 0.8 to >1 relative density range compared to CPUE in the 0.3 to 0.5 range (both dark blue and easily distinguished by the shape in the coarse resolution figures.

Response 2: We use different symbols (squares and circles) to distinguish prediction and observation. For better visualization of the results, observed CPUEs were shown from green to red, and the predictions were shown from light to dark blue. We can compare high values of observed and predicted CPUE by blue and red in the plots.

Point 3: I would like the authors to discuss the differences in SST between the two models before the conclusions since I am curious about these differences and think other readers will be as well. I am glad they were mentioned in the conclusions but think perhaps that should be discussed earlier as well.

Response 3: Thanks for the suggestions. A paragraph to address this issue has been added in the discussion (Line 848-858).

“This difference was as expected because SST is highly correlated with latitude, and the HSI approach cannot deal well with the collinearity between these two effects. By contrast, the latitude effect, the longitude effect, and the interaction between the two were included in the GAM analysis, which improves the identification of SST’s effect from the spatial factors.”

Point 4: The manuscript is a bit wordy and some of the description of the model results are repeated too frequently e.g. there is no need to repeatedly say that there are two different spatial resolutions used .... but this was a minor concern. My other minor comments and suggestions are included in the attached and edited manuscript.

Response 4: The manuscript has been professionally edited. Thanks for the comments. We have revised the paper according to all the suggestions.

Reviewer 2 Report

The paper entitled "Modeling the spatial distribution of swordfish (Xiphias gladius) using fishery and remote sensing data with consideration on approach and resolution" is a well written important paper. I recommend that the work is suitable to be published in Remote Sensing journal, with the following minor comments (provided the authors address them).

Page 2, Line 59-60: However, similar investigation ......... in the Pacific Ocean". The sentence has grammatical issues. Please rephrase. Page 5, Line 183: The values "26.2%" and "29.4%" for SST based explained deviance. I believe it's a fraction between the values of SST and the sum of all deviance explained (Table 1). It seems the values are a bit off for corse grid it is ~29.87% and for the fine grid it is 26.9% Page 5, Line 184: Also the figure 37.7% and 46.0% looks a little off. If these values are from the table 1, then can the authors explain how these values are calculated. Page 5, 191-192: "In the analysis of ......... (Figure 3)." - Please rephrase this sentence, as, in it's current form I do not understand what it says. Page 11, Line 295-298: "Without sufficient data ......... habitat ranges [22]" - Change to: "Without sufficient fishery data from the Southwestern Pacific Ocean and the support of ..............." Page 13: Conclusion is a bit generalized and lacks clarity. Please include your results/outcomes from this study and highlight them with some quantitative data.

Author Response

Point 1: Page 2, Line 59-60: However, similar investigation ......... in the Pacific Ocean". The sentence has grammatical issues. Please rephrase.

Response 1: Thanks. This sentence has been revised as follows (Line 96-97).

However, similar investigations that cover a large geographical area are lacking for swordfish in the Pacific Ocean.

Point 2: Page 5, Line 183: The values "26.2%" and "29.4%" for SST based explained deviance. I believe it's a fraction between the values of SST and the sum of all deviance explained (Table 1). It seems the values are a bit off for corse grid it is ~29.87% and for the fine grid it is 26.9%. Page 5, Line 184: Also the figure 37.7% and 46.0% looks a little off. If these values are from the table 1, then can the authors explain how these values are calculated.

Response 2: The values have been updated for the final best fitted models (Line 495-498).

The most important environmental factor affecting the swordfish catch rates was SST, which accounted for 29.9% and 26.9% of the explained deviance for the coarse and fine resolutions, respectively. SSS, SSH, and CHL explained 46.7% and 38.7% of the explained deviance for the two final selected GAMs for the coarse and fine resolutions, respectively (Table 1).

Point 3: Page 5, 191-192: "In the analysis of ......... (Figure 3)." - Please rephrase this sentence, as, in it's current form I do not understand what it says.

Response 3: The sentence has been rephrased as follows (Line 504-505).

When the fine grid was used, high relative densities of swordfish were highly related to the change of lunar phase (Figure 3).

Point 4: Page 11, Line 295-298: "Without sufficient data ......... habitat ranges [22]" - Change to: "Without sufficient fishery data from the Southwestern Pacific Ocean and the support of ..............."

Response 4: Thanks. The sentence has been revised as follows (Line 771-772).

Without sufficient fishery data from the southwestern Pacific Ocean and the support of the spatial determinant (latitude and longitude) in the GAM

Point 5: Page 13: Conclusion is a bit generalized and lacks clarity. Please include your results/outcomes from this study and highlight them with some quantitative data.

Response 5: The conclusion has been revised to emphasize the importance on exploring the modelling approaches and how the environmental effects are treated, and the data resolution is a minor issue for this case (Line 869-878).

Reviewer 3 Report

Dear Authors,

first, I don't think that you selected a proper journal. You use few remote sensing indices, but you don't have any input to the remote sensing society, you do not propose any new methods and applications. You should think about another journal, e.g. Fishes, Sustainability or Water, but you need to improve the manuscript because all parts are significantly too general. 

You do not present a proper theoretical background and current solutions of your topic. You need to add more details oriented on input data, accuracy assessment, used algorithms and validation of used methods.

Then you need to analyse the most important results and to discuss/compare with proper references.

Author Response

We have revised the manuscript according the Editor's suggestions. The title of section 2.4 has been changed to Model validation and accuracy assessment. We have highlighted the major finding in the conclusion and improve the MS including the English being professionally edited.